# Carbon Allocation to Leaves and Its Controlling Factors and Impacts on Gross Primary Productivity in Forest Ecosystems of Northeast China

**Zhiru Li [1], Quan Lai [1,2,*], Yuhai Bao [1,2], Bilige Sude [3,4], Zhengyi Bao [5] and Xinyi Liu [1]**

[1] College of Geographical Science, Inner Mongolia Normal University, Hohhot 010022, China; 20214019005@mails.imnu.edu.cn (Z.L.)

[2] Inner Mongolia Key Laboratory of Remote Sensing and Geographic Information Systems, Inner Mongolia Normal University, Hohhot 010022, China

[3] Information Center, Inner Mongolia Normal University, Hohhot 010022, China

[4] Department of Geography, School of Arts and Sciences, National University of Mongolia, Ulaanbaatar 14201, Mongolia

[5] College of Computer Science and Technology, Inner Mongolian Normal University, Hohhot 010022, China

[*] Correspondence: laiquan@imnu.edu.cn; Tel.: +86-186-8601-6069

**Abstract:** Carbon allocation in forest ecosystems is essential for the optimization of growth. However, remote-sensing-based research on the estimation of carbon allocation in forests is inadequate. This article considers forests in northeastern China as the research area and uses leaf area index (LAI) data combined with random forest and structural equation modelling methods to study the spatiotemporal distribution characteristics and driving factors of carbon allocation to leaves (ΔLAI) in deciduous broad-leaved forests (DBF), deciduous coniferous forests (DNF), and mixed forests (MF) during the green-up period (GUP) at a monthly scale during April, May, June, and July from 2001 to 2021, and clarifies the impact of leaf carbon allocation on gross primary productivity (GPP). The ΔLAI was the highest in DBF in April and in DNF and MF in May. The ΔLAI in April with an increasing trend year by year in DBF and MF, and the ΔLAI in May with an increasing trend in DNF. Among all the direct and indirect relationships that affect ΔLAI, temperature (TEM) has the highest path coefficient for DBF's ΔLAI in April (−1.213) and the start of the season (SOS) has the highest path coefficient for DNF (−1.186) and MF (0.815). ΔLAI in the GUP has a significant positive impact on the GPP. In the MF, the higher ΔLAI in May was most conducive to an increase in GPP. During the critical period, that is April and May, carbon allocation to leaves effectively improves the carbon sequestration capacity of forestland. This information is of great value for the development and validation of terrestrial ecosystem models.

**Keywords:** carbon allocation to leaves; phenology; climate; structural equation model; random forest

## 1. Introduction

The $CO_2$ present in the Earth's atmosphere enters the biosphere through the photosynthesis of vegetation [1]. Forests, the world's largest carbon sink, play a crucial role in the global carbon cycle. For decades, the world's existing forests have been stable carbon sinks, sequestering approximately 30% of the world's total anthropogenic carbon emissions [2]. Assimilated carbon is transferred between photosynthetic organs (leaves) and non-photosynthetic organs (wood and roots) for different functions [3,4]. Forests maintain their internal function and structural stability by regulating the proportion of carbon allocated to different physiological structures, and this phenomenon is termed carbon allocation [5]. This is an important physiological process for optimizing forest growth, wherein forests interact with ecosystems through carbon allocation processes, thereby affecting the carbon balance in them, even on a large scale [6,7]. However, the dominant factors that

cause these physiological changes and the potential driving mechanisms of plant carbon allocation strategies have not been adequately explored.

Current research on carbon allocation to leaves has made significant progress at the canopy scale [8,9]. Carbon allocation is estimated mainly by measuring physiological characteristics such as dry and wet weights of leaves, breast height radius of the tree, and tree age at different growth nodes. However, collecting information on these in forests over a large area is time-consuming, labour-intensive, and infeasible. The widespread application of satellite remote sensing technology in the field of ecology provides a new method for studying the carbon distribution process in leaves over a larger area [10,11]. However, the drivers of carbon allocation to leaves in forests of northeast China and their contributions to terrestrial carbon sinks have not yet received attention.

Carbon allocation to leaves in trees is driven by both internal resource constraints and multiple external factors [12]. Under the combined effects of these factors, the growth environment of trees undergoes dynamic changes. At a large regional scale, the driving factors of the processes of carbon allocation to leaves in the entire forest ecosystem are complex, and factors such as hydrothermal conditions and phenology may become the main driving factors of leaf carbon allocation [13]. Currently, research is conducted from the perspective of monitoring carbon allocation strategies of different physiological tissue structures in trees during their growth at the microscale. However, these studies lack the ability to explore the main controlling factors of carbon allocation to leaves in different forest ecosystems at the macro scale and do not address the relative influence of size and action paths between different controlling factors. The impacts of carbon allocation strategies in forest ecosystems should not be ignored.

Some studies have found that trees adapt to environmental stress through dynamic physiological adjustment mechanisms, thereby reducing negative impacts and maximizing carbon sequestration [8]. Gross primary productivity (GPP) is an important parameter for studying global climate, carbon cycle change, and global ecosystems. However, the extent of the effects of changes in carbon allocation to leaves remains unknown. Current research on the factors affecting GPP mainly considers the role of climatic factors, whereas the indirect impact of forest adaptation strategies (carbon allocation) on the ecosystem has been insufficiently studied, although this adjustment strategy within trees may be greater than the direct impact of climate [14].

Forests in northeast China are located in a temperate monsoon climate zone. These forests are the largest and most complete virgin Korean pine forests in Asia and have high carbon sequestration potential [15]. They can promote rainwater circulation in local areas, improve climatic conditions, and even regulate the global climate. However, studies on carbon allocation in forests in northeast China have been conducted at a small regional level. There is a need for research at the ecosystem and regional levels. Therefore, this study aimed to reveal the spatiotemporal distribution characteristics and potential driving factors of carbon allocation in forest leaves of the forest of northeast China, and based on this, explain the impact of leaf carbon allocation on GPP. Exploring the spatiotemporal distribution and driving factors of carbon allocation to leaves in different forests in northeast China is of great significance as it can provide valuable information for regional ecological management and formulation of policies.

## 2. Materials and Methods

### 2.1. Overview of the Study Area and Data Sources

#### 2.1.1. Study Area

The forestland area of northeast China was chosen as the research area to explore the spatiotemporal distribution of carbon allocation to leaves and its driving factors. This study region lies between 115°50′ E–134°25′ E and 38°24′ N–53°23′ N, encompasses Hulunbuir City, Xing'an League, Chifeng City, and Tongliao City in the Inner Mongolia Autonomous Region and Heilongjiang, Jilin, and Liaoning Provinces, and covers a total area of 1.15 million km² (Figure 1a). This region has the largest forest coverage in northern

China and consists of deciduous broad-leaved forest (DBF), deciduous needleleaf forest (DNF), and mixed forest (MF).

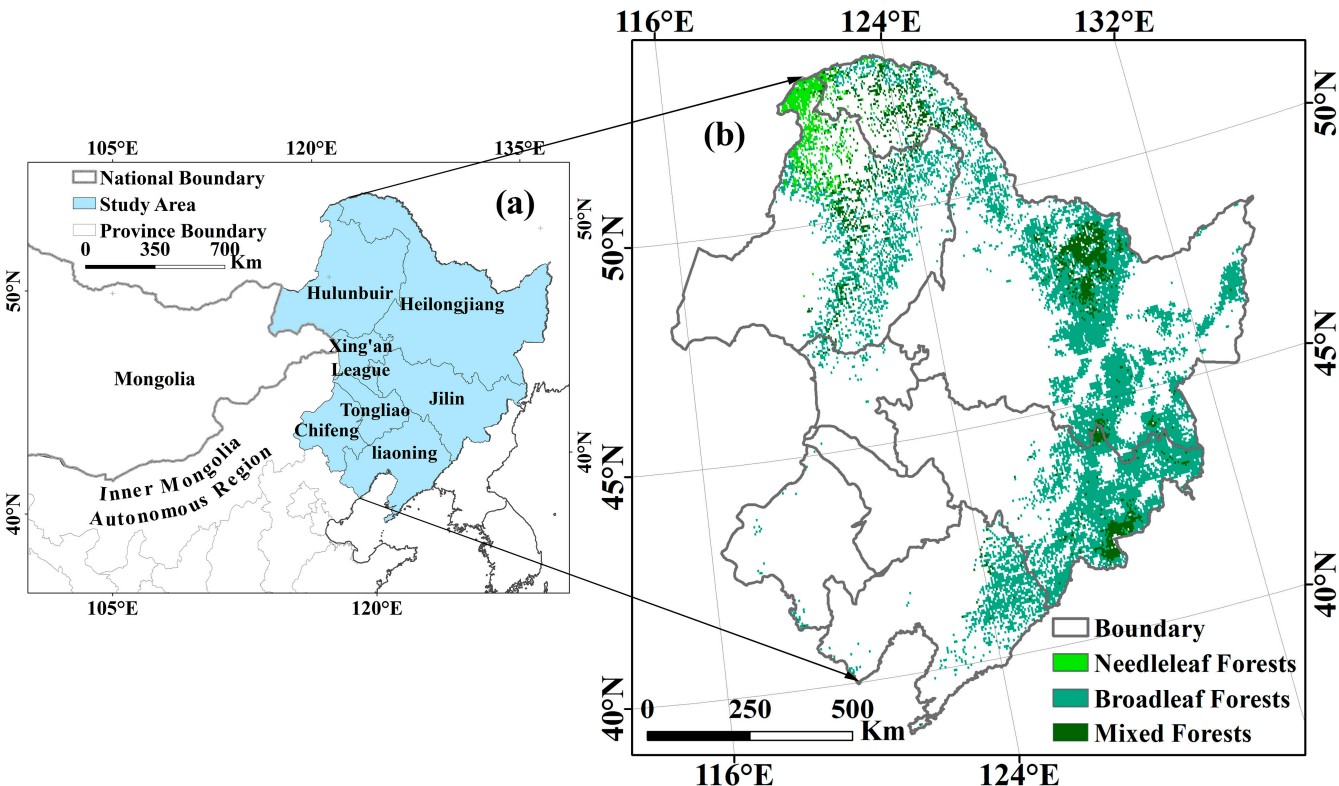

**Figure 1.** (**a**) Location of the study area. (**b**) Distribution of forest types in the study area.

### 2.1.2. Remote Sensing Data

This study used remote sensing data on leaf-area index (LAI) based on GLASS products from the National Science and Technology Basic Conditions Platform–National Earth System Science Data Center (http://www.geodata.cn, accessed on 13 July 2023) LAI-MODIS dataset, with a spatial resolution of 0.05° and a time resolution of 8 d for the period January 2001 to December 2021. This LAI dataset reasonably removed noise and had good temporal stability, which compensated for the lack of measured values. The maximum value composite method was used to synthesize data at a monthly scale, which was then used to calculate the monthly increase in LAI (ΔLAI), and annual-scale data were synthesized based on these. The difference between the annual maximum LAI and the LAI in the month before the rejuvenation period is defined as the ΔLAI within the green-up period (GUP). Based on the land cover classification of MCD12C1 v006, the DBF, DNF, and MF types of forested land areas were selected for this study (https://lpdaac.usgs.gov/products/mcd12c1v006/, accessed on 18 August 2023).

Forests in northeast China were analyzed using 0.1° monthly scale temperature (TEM), precipitation (PRE), and solar radiation (SR) data obtained from ERA-5 Land products provided by the European Center for Medium-Range Weather Forecasts (ECMWF) (https://cds.climate.copernicus.eu/, accessed on 17 July 2023). The impact of meteorological factors on carbon distribution in leaves of forests in the region was evaluated.

This study used the phenology product MCD12Q2 V6, which was calculated using data from the Moderate Resolution Imaging Spectroradiometer (MODIS) (https://lpdaac.usgs.gov, accessed on 13 July 2023). We defined the GUP of leaves as the time between the start of the season (SOS) and the peak of the season (POS). SOS is defined as the date at which the value of the enhanced vegetation index value exceeds 15% for the first time.

This study used the global solar-induced chlorophyll fluorescence gross primary production (GOSIF GPP) data product derived from the research results of the Global

Ecology Group of Li and Xiao (https://globalecology.unh.edu, accessed on 21 June 2022). This dataset was based on the global SIF product of OCO-2 and the linear relationship between SIF and GPP [16]. We used the 8-day-scale GOSIF GPP from 2001 to 2021 to calculate the GPP within the annual growing season to study the impacts of ΔLAI.

*2.2. Methods*

2.2.1. Estimation of Carbon Allocation to Leaves

We used the ΔLAI measured in each month and throughout the GUP to represent the net allocation of carbon to the leaves (hereafter referred to as leaf carbon), which is the difference between the total carbon allocation to leaves and leaf respiration. The ΔLAI within the GUP is defined as the annual maximum LAI minus the LAI of the month before the SOS. The monthly ΔLAI is calculated as follows:

$$\Delta LAI = LAI_t - LAI_{t-1} \tag{1}$$

As leaf growth is irreversible during the greening period [17], the ΔLAI should always be positive during this period. Therefore, we discarded pixels with ΔLAI < 0 from the analysis.

2.2.2. Forecast of Future Trends

First, we used a linear regression model to determine the interannual trend of ΔLAI during the entire study period, and the regression slope represents the trend rate [18]:

$$(\Delta LAI)_i = a_i * (\text{Time}) + b_i \tag{2}$$

where *Time* is the number of years from 2000 to 2017 for pixel i, $a_i$ is the time trend of ΔLAI of pixel i, and $b_i$ is the intercept of pixel i.

Second, to analyze the persistence of changes in ΔLAI, the Hurst index was used, which was calculated using the rescaled range method (R/S). The calculation method is as follows.

Suppose there is a time series {ξ(t)}, t = 1, 2, 3, $\cdots$, any positive integer $\tau \geq 1$, and a mean sequence exists:

$$<\xi>_\tau = \frac{1}{\tau}\sum_{t=1}^{t} t = \xi(t), t = 1, 2, 3, \cdots \tag{3}$$

From this, the cumulative dispersion is:

$$X(t, \tau) = \sum_{u=1}^{t}[\xi(t) - <\xi>_\tau], 1 \leq t \leq \tau \tag{4}$$

The calculation formula of range R is:

$$R(\tau) = \max X(x, \tau) - \min X(x, \tau), \ \tau = 1, 2, 3 \ldots 1 \leq t \leq \tau \tag{5}$$

R, S, and τ have the following relationship:

$$\frac{R_\tau}{S_\tau} = c\tau^H \tag{6}$$

$$\lg\left(\frac{R(\tau)}{S(\tau)}\right) = \lg c + H\lg\tau \tag{7}$$

$\frac{R(\tau)}{S(\tau)}$ is the rescaled range; H is the Hurst index; c is a constant. Using Equation (7), the estimated value of the Hurst index can be obtained using the least-squares regression method. By superimposing the Hurst index and the slope value, the changing trend of ΔLAI can be classified into 5 categories as shown in Table 1.

**Table 1.** Classification of future change trends of ΔLAI.

| Hurst | Slope | Future Trends |
|:---:|:---:|:---:|
| >0.5 | >0 | Continuous increase |
| | <0 | Continuous decrease |
| <0.5 | >0 | From increase to decrease |
| | <0 | From decrease to increase |
| =0.5 | | Unpredictable |

### 2.2.3. Statistical Analyses

(1)    Pearson correlation analysis

Pearson's correlation analysis was used to determine the relationship between forest carbon allocation to leaves and the GPP at different timescales [19]. Statistical significance was set at $p < 0.05$. In the northern hemisphere, spring includes March, April, and May; summer includes June, July, and August; autumn includes September, October, and November; and winter includes December, January, and February. However, owing to the special drought conditions in Inner Mongolia, most areas did not turn green in March, and the forests entered the withering and yellowing stages in November. Therefore, this study only examined the correlation between carbon allocation and the GPP from April to October for the corresponding month.

(2)    Random forest (RF)

This study used the random forest (RF) method to analyze the importance of six factors on carbon allocation to leaves in different time periods before the occurrence of the growth peak from 2001 to 2021. The RF method is both a supervised machine learning algorithm and an ensemble algorithm that builds and combines multiple decision trees to create a "forest" to obtain more accurate and stable results than that obtained using a single tree [20]. RF extracts observation samples from the training set, replaces them with bootstrap samples, randomly selects a feature subset in the tree model to form a collection of tree models, and combines the results by voting (classification) or averaging (regression) [21]. Therefore, RF can reduce the overfitting of individual trees and correlations between trees. In this study, RF was used for the regression, and it was implemented using the "randomForest" package [22] in R.

(3)    Structural equation model

To further analyze the direct and indirect effects of environmental factors on forest carbon allocation to leaves at different time scales, the "lavaan" package in R version 4.1.3 software was used to construct the structural equation model (SEM) [23,24]. The SEM adopts the maximum likelihood estimation method. The chi-square ($\chi^2$) test was used to evaluate the fitness of the model, that is, whether $p > 0.05$, the standardized root-mean-square residual (RMSEA) < 0.05, and the relative fit index (SRMR) < 0.05, and goodness-of-fit index (GFI) > 0.95, to evaluate the degree of model fit [25].

## 3. Results

### 3.1. Spatiotemporal Distribution of Carbon Allocation to Leaves

#### 3.1.1. Interannual Variation Trend of Carbon Allocation to Leaves

This study investigated the interannual change trends in carbon allocation to leaves at different growth stages from 2001 to 2021 and in the future on a pixel-by-pixel basis (Figure 2). Research has shown that the ΔLAI within the GUP will continue to increase in 18.9% of the area, and approximately 1% of the area will continue to decrease (Figure 2a). In addition, the changing trend of monthly ΔLAI (LAI of the current month minus the ΔLAI of the previous month) indicates the trend of leaf carbon accumulation on a monthly scale over many years, with varying trends in different months. An area of 34.8% that had a continuously increasing trend in ΔLAI in April is mainly distributed in the DBF in the southeast. A quantity of 0.6% of pixels with a continuously decreasing trend is

mainly concentrated in the northern DNF area (Figure 2b). Areas with a continuously increasing trend in April showed a continuously decreasing trend in May, and 10.7% of the areas in the northwest showed a continuously increasing trend in ΔLAI in May (Figure 2c). Less than 5% of the area showed a sustained increasing or decreasing trend in June (Figure 2d). According to the change characteristics of ΔLAI from July 2001 to July 2021, an unpredictable trend was observed for July in almost the entire study area (Figure 2e).

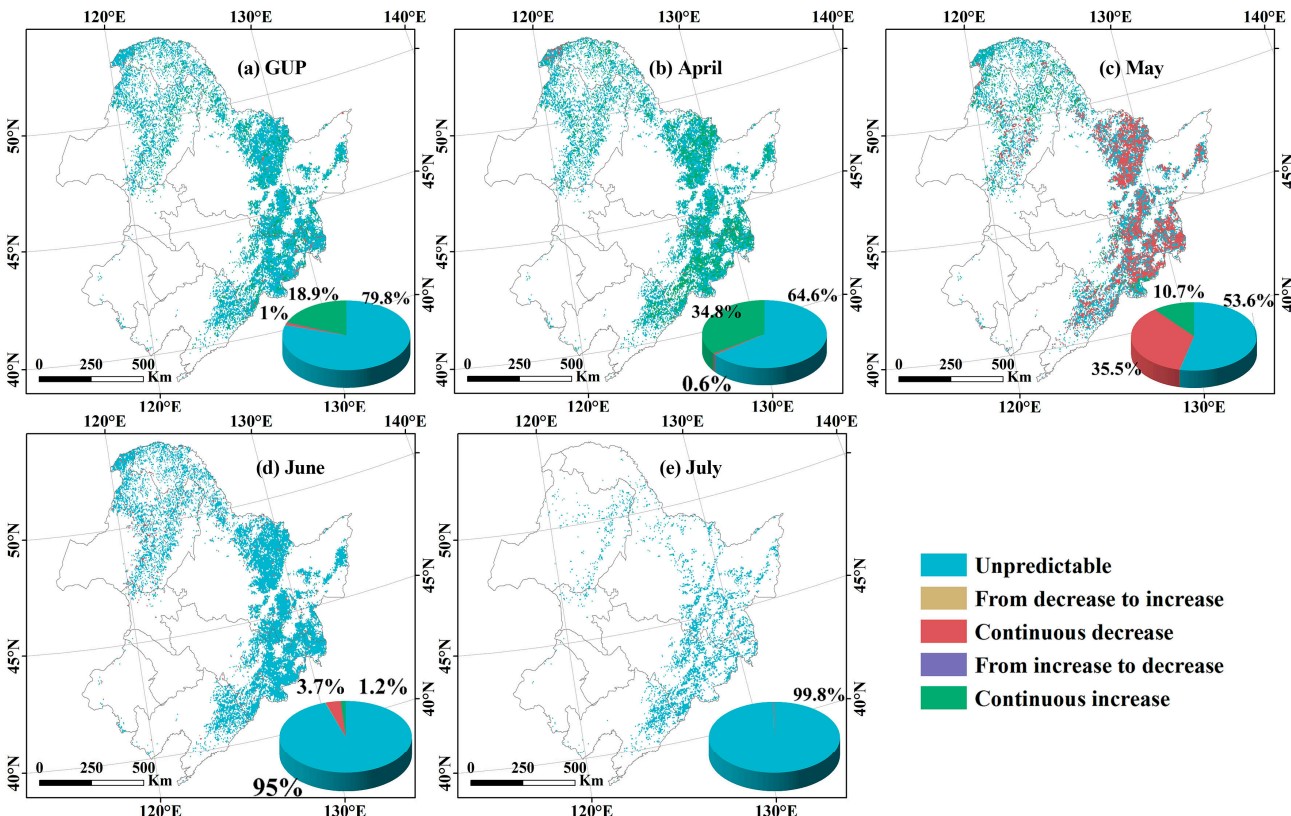

**Figure 2.** Change trends of ΔLAI in the entire leaf during (**a**) green-up period (GUP) (GUP ΔLAI is defined as the difference between the annual maximum LAI and the LAI one month before the start of the growing season) and during (**b**) April, (**c**) May, (**d**) June, and (**e**) July (monthly ΔLAI is defined as the difference between the current month's LAI and the previous month's LAI) from 2001 to 2020 and in the future.

The spatial heterogeneity of ΔLAI trends in different months within the forest GUP in northeast China is related to the physiological attributes of different forest vegetation types. This is manifested in the fact that the main months of leaf growth differ for different forest types. The ΔLAI of different forest types at different stages was also significant (Figure 3). There were significant differences in the ΔLAI among the different forest types within the entire GUP. It was the highest for DBF, followed by MF and DNF. In terms of the change trends, DBF and MF were similar (slope = 0.19). The interannual rate of increase in the DNF is significantly lower at 0.098 (Figure 3a). The ΔLAI trends in April exhibited different patterns. The ΔLAI of all forest land types was the lowest from 2003 to 2005. After 2005, all forest types showed an increasing trend, with varying degrees of fluctuation. The highest rate of increase was observed for DBF and MF at 0.35 and 0.36, and for the DNF, with the lowest rate of change, it was 0.09 between 2001 and 2021(Figure 3b). The interannual fluctuation range of ΔLAI was the highest in May, and it was more different for the three forest types than those in April and GUP. The ΔLAI in May of DBF and MF decreases year by year at rates of 0.11 and 0.08. However, DNF showed a trend of insignificant increase at the rate of 0.09. Additionally, the ΔLAI of MF was the highest before 2011, and the ΔLAI

of DNF was the highest after that (Figure 3c). The ΔLAI of DBF was the highest in June, and those of DNF and MF were relatively low. The ΔLAI of the three forest types in June showed a consistent downward trend from 2001 to 2021 (Figure 3d). The difference in ΔLAI among the three types of forests was obvious in July, with DBF being the highest, followed by MF and DNF. MF and DNF showed insignificant increasing trends, whereas DBF showed no obvious trend (Figure 3e).

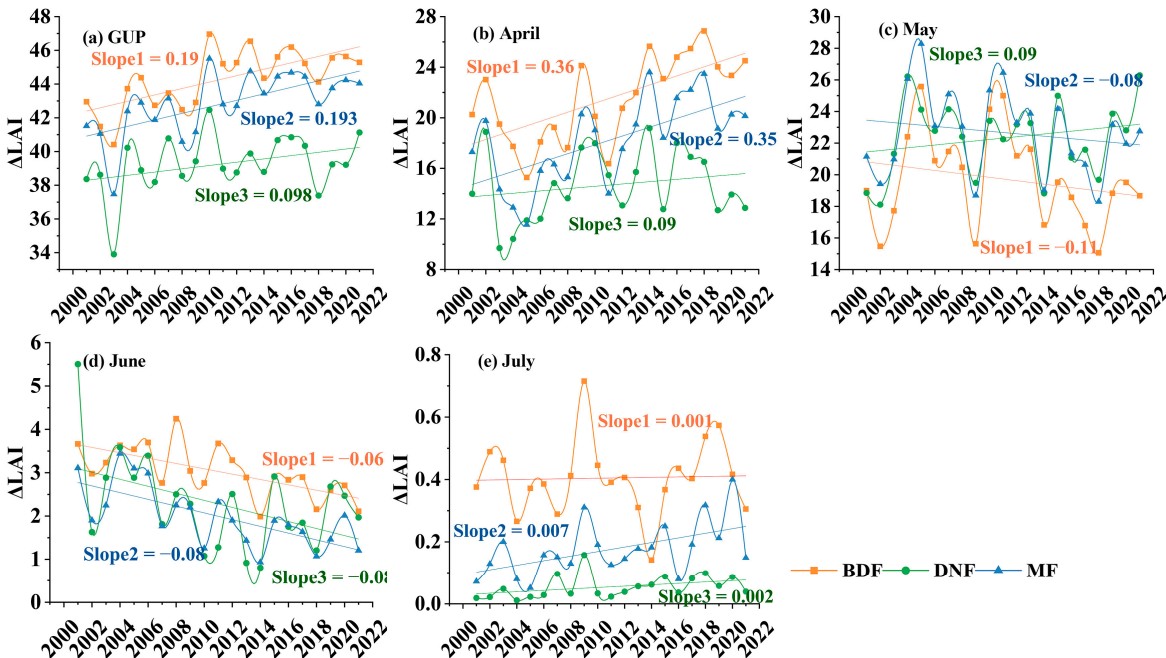

**Figure 3.** Trends of the entire leaf green-up period (GUP) and monthly ΔLAI in different forest types from 2001 to 2021. (**a**) The trend of ΔLAI across the GUP (ΔLAI is defined as the difference between the annual maximum LAI and LAI one month before the start of the growing season). ΔLAI trends in (**b**) April, (**c**) May, (**d**) June and (**e**) July (monthly ΔLAI is defined as the difference between the current month's LAI and the previous month's LAI).

### 3.1.2. Spatial Distribution of Carbon Allocation to Leaves

Carbon allocation to leaves in forests showed obvious spatial heterogeneity at different growth stages (Figure 4). Within the entire GUP, ΔLAI has an overall spatial characteristic of low in the northwest and high in the southeast. This spatial difference was more significant in April, whereas the spatial distribution in May was completely the opposite of that in April (Figure 5b,c). When forest growth reaches its peak in June, the ΔLAI of the entire region decreases significantly compared with the previous period (Figure 5d). As pixels with increased carbon allocation to leaves were only counted (i.e., areas where ΔLAI > 0), Figure 5e shows that ΔLAI no longer increases in >70% of the area in July, and the increase in other areas is <1 m² m⁻², which shows that almost all forests would have completed leaf development by July.

Differences in the spatial distribution of carbon allocation to leaves among different forest types were observed. Within the GUP, DBF and MF have a higher ΔLAI than DNF, and ΔLAI peaks occur in April and May. However, the months in which the increase in ΔLAI occurred for different forest types varied and showed an obvious longitudinal zonality. Because of the latitudinal advantage, the main month of carbon allocation to leaves in the southeastern region was April and it was dominated by DBF, whereas in the northwestern region, it was dominated by DNF. The carbon allocation to leaves was the highest in May due to differences in longitude (Figure 4b,c).

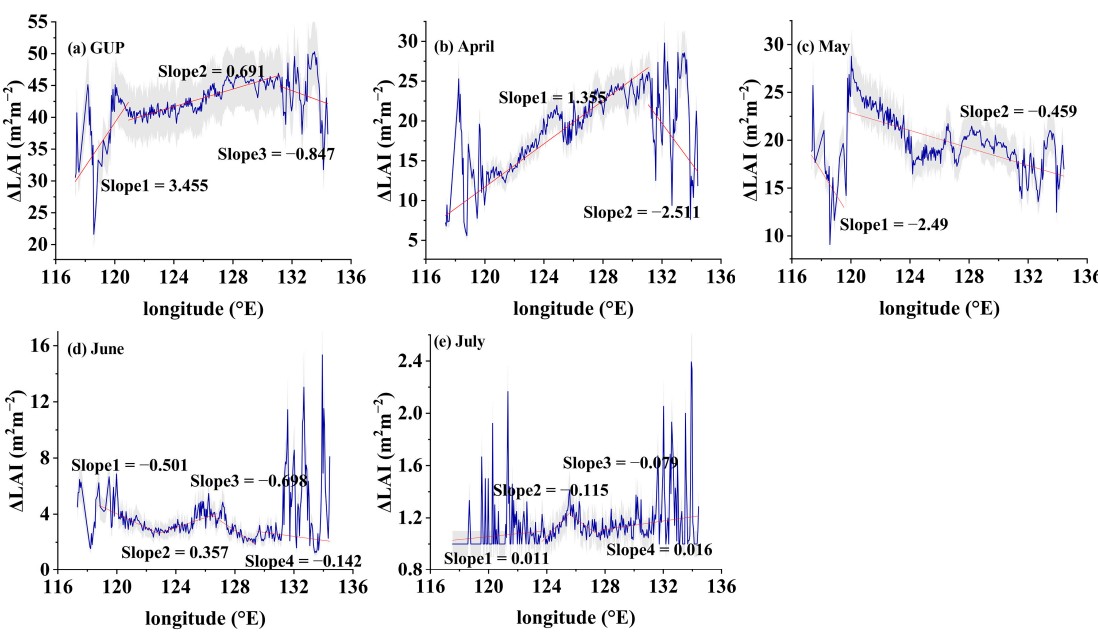

**Figure 4.** The spatial distribution of eigenvalues of ΔLAI in the horizontal direction. (**a**) The entire GUP ΔLAI (ΔLAI is defined as the difference between the annual maximum LAI and LAI in the month before the start of the growing season). (**b**) April, (**c**) May, (**d**) June, and (**e**) July ΔLAI (Monthly ΔLAI is defined as the longitudinal zonal variation pattern between the LAI of the current month and the LAI of the previous month).

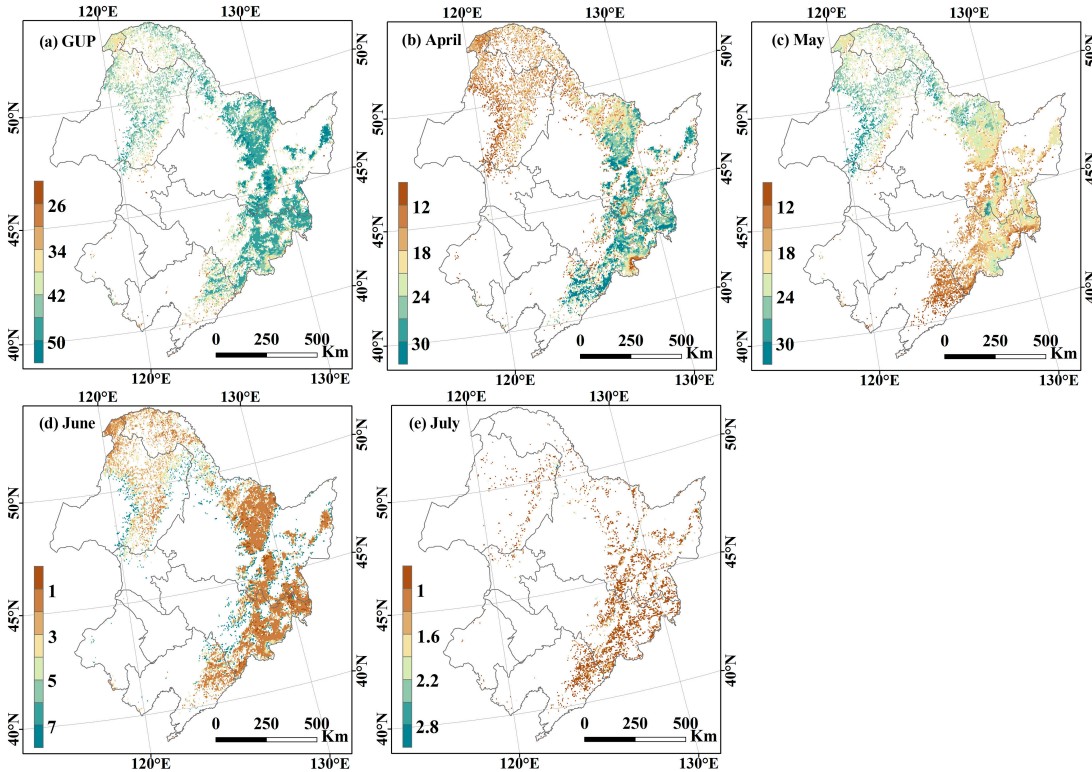

**Figure 5.** The spatial distribution of ΔLAI of the entire green-up period (GUP) and every month from 2001 to 2021. (**a**) The entire GUP ΔLAI (ΔLAI is defined as the difference between the annual maximum LAI and LAI in the month before the start of the growing season). (**b**) Multi-year average for April, (**c**) May, (**d**) June, and (**e**) July ΔLAI (Monthly ΔLAI is defined as the difference between the current month's LAI and the previous month's LAI).

### 3.2. Analysis of Driving Factors of Carbon Allocation to Leaves

As shown in Figure 2e, leaf development is completed in July, and hence, we comprehensively analyzed the importance of TEM, PRE, SR, and SOS to carbon allocation to leaves (ΔLAI) from April to June (Figure 6). There were significant differences in the main influencing factors in different forest types. Within the GUP, all influencing factors had the greatest impact on carbon allocation to leaves in DBF, followed by MF and DNF. For DBF, the impacts of SOS and SR were higher than those of other factors. In MF, the impact of SOS was significant, and those of other factors were low. In DNF, except for the low importance of SR, the other factors were similar (Figure 6a). In April, SOS had the highest impact on all forestlands. For DBF and MF, the main influencing factors of ΔLAI were SOS and TEM, followed by SR, whereas PRE was the lowest. All factors had a greater impact on DBF than on MF. For DNF, except for SOS, the importance of the other factors was similar. It is worth noting that the impact of PRE on DNF was greater than that of other forestlands, whereas the impact of TEM was lower than that of other forestlands, and the impacts of SOS and SR were intermediate (Figure 6b). In May, the pattern remained almost the same as that observed in April. The impact of SOS remained high only in DBF, and its impact on other forestlands was significantly lower than that in April. The impact of TEM was lower than that in April. In DNF, except for SOS, the influence of the other three factors increased compared to April, and PRE exceeded that of TEM. This was similar to that in April in the MF (Figure 6c). SOS has the most significant lag effect on ΔLAI in June. Compared to May, the impact of all factors was significantly reduced in June. Only SOS in MF had a greater impact in June than in May, and its importance was higher than those of other climatic factors in the same time period. This phenomenon was most evident in MF, followed by DBF, with DNF having the lowest value. Other factors also confirmed this pattern (Figure 6d).

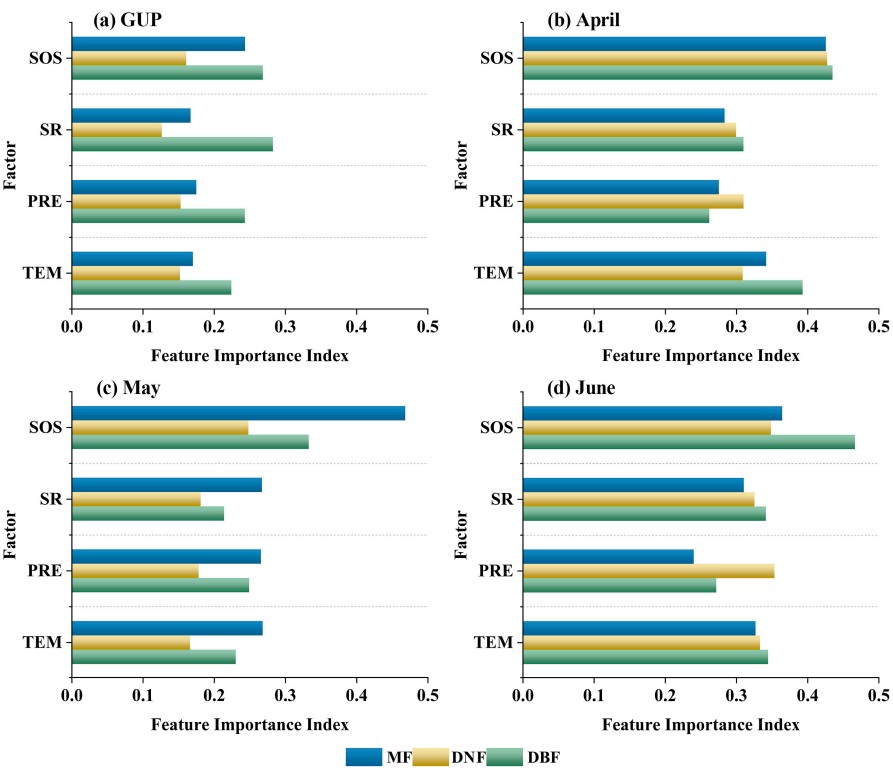

**Figure 6.** The feature importance indexes of deciduous broadleaf forest (DBF), deciduous needleleaf forest (DNF), and mixed forest (MF) in northeast China from 2000 to 2021 based on random forests analysis in (**a**) the entire GUP (ΔLAI is defined as the difference between the annual maximum LAI and LAI in the month before the start of the growing season), (**b**) April, (**c**) May, and (**d**) June (months ΔLAI is defined as the difference between the current month's LAI and the previous month's LAI).

By constructing structural equations, we further explored the direct and indirect effects of the quantified TEM, PRE, SR, and SOS on carbon allocation to leaves (ΔLAI) (Figure 7). Owing to multicollinearity in some indicators, this study only selected representative indicators to construct the overall fit of the SEM to the standard. Obvious differences in the driving factors of ΔLAI of different forest types in different time periods were observed. The indirect effect coefficient of the meteorological factors on ΔLAI through SOS was higher than that of the direct effect coefficient.

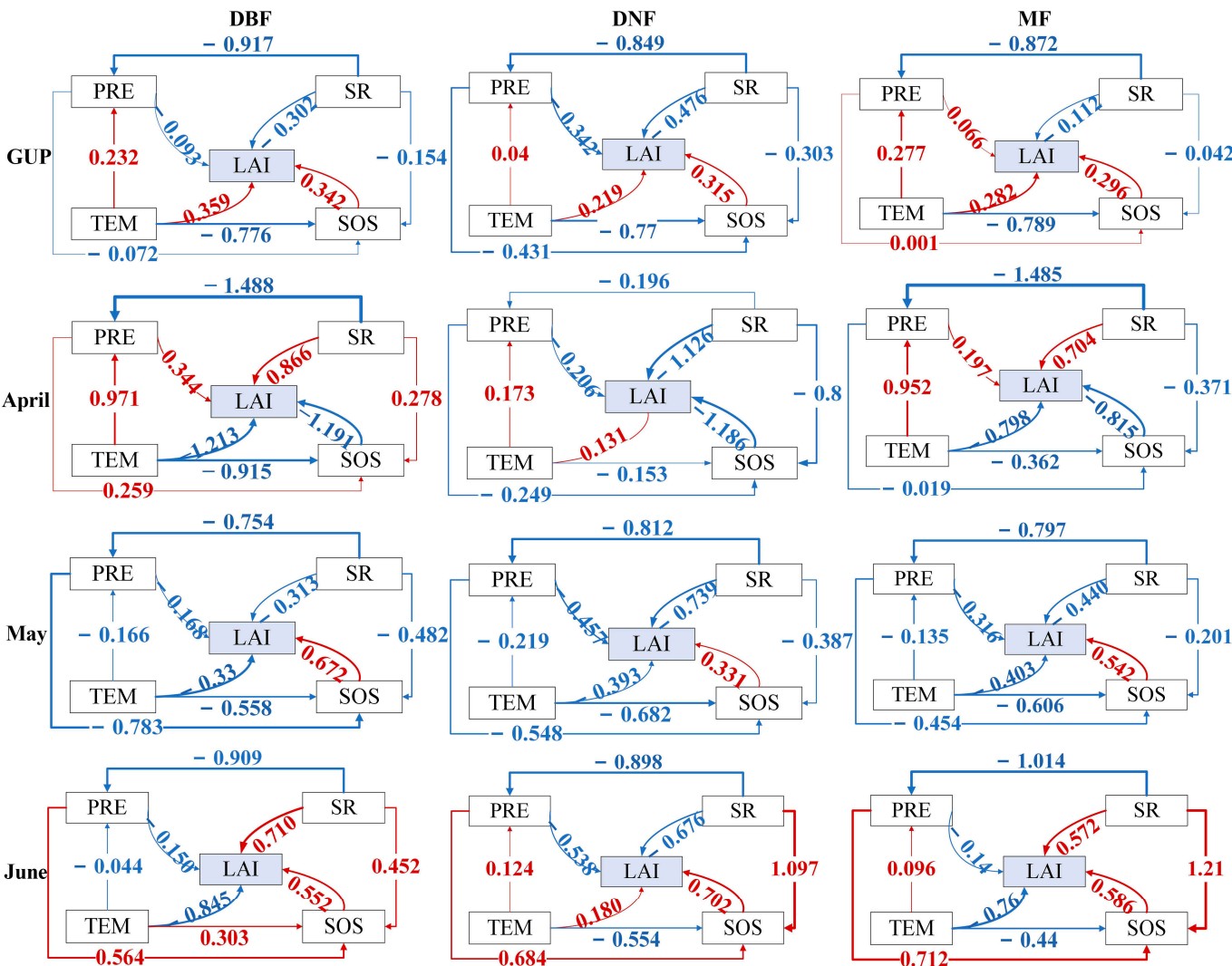

**Figure 7.** The structural equation model (SEM) examines the relationship between carbon allocation to leaves (ΔLAI), phenological greening stage (SOS), air temperature (TEM), precipitation (PRE), and solar radiation (SR). The red and blue lines represent positive and negative coefficients respectively; the thickness of the arrow represents the size of the standardized path coefficient.

In GUP, all factors have similar driving mechanisms for ΔLAI in DBF and DNF, with positive path coefficients for TEM and SOS, and negative effects on PRE and SR. However, this differs in DBF, wherein TEM has the highest positive path coefficient (0.359), and in DNF, wherein SOS has the highest coefficient (0.315). Among the negative effect coefficients, both DBF and DNF have the highest SR, with values of 0.302 and 0.476, respectively. In MF, except for SR, all are positive path coefficients, with SOS being the maximum value (0.296).

In April, SOS had high path coefficients for all forest types, and in descending order as follows: DBF (−1.191) > DNF (−1.186) > MF (−0.815). Specifically, the factors with the highest positive and negative path coefficients in DBF are SR (0.866) and TEM (−1.213), respectively. In DNF and MF, SOS had the highest negative path coefficient, while PRE (0.206) and SR (0.704) had the highest positive path coefficient in the two woodlands, respectively.

In May, all direct path coefficients for ΔLAI are negative except SOS. The relationship between SOS and ΔLAI shows a pattern opposite to that in April and is the only positive direct driver, and in descending order as follows: DBF (0.672) > MF (0.542) > DNF (0.331). The highest negative path coefficients are TEM (−0.33) and SR (−0.44) in DBF and MF respectively. In DNF, SR has the highest negative path coefficient (−0.739).

In June, the path coefficients of SOS in all forest lands were positive, with DNF being the highest (0.782), followed by MF (0.586), and DBF (0.552). In DBF, the positive effect of SR exceeds that of SOS and becomes the highest positive driving force with a path coefficient of 0.71, and TEM has the highest negative effect coefficient (−0.845). The highest positive coefficient of ΔLAI in DNF and MF is SOS. The difference is that SR is the only negative path coefficient in DNF (−0.676), while TEM has the highest negative path coefficient in MF (−0.76).

### 3.3. Effects of Carbon Allocation to Leaves on Gross Primary Productivity

Carbon allocation to leaves (ΔLAI) at different growth stages (April to June) has varying degrees of impact on the GPP (Figure 8). Overall, increased carbon allocation to leaves during the GUP of the forest was most beneficial for GPP accumulation in September. There is a lag effect between ΔLAI and GPP in different months of the forest. The ΔLAI in April had the strongest positive correlation with GPP in May (Figure 8b, R = 0.801), indicating a lag time of 1 month. From June to October, this positive correlation gradually weakened, and even became negative. The ΔLAI in May showed a strong positive correlation with the GPP in all months except May. The promotion effect of the increase in ΔLAI on the GPP in June was significantly lower than that in May, and it had the strongest lag effect, with a lag time of two months, on the GPP in August (Figure 8m, R = 0.218).

The correlation of ΔLAI and GPP in the GUP for different forest types was similar; however, it varied for different months. For MF, the increase in ΔLAI in April has a positive effect on the GPP from April to June, but it has an inhibitory effect on the GPP in July and subsequent months. The ΔLAI in May is positively correlated with the GPP in July and subsequent months. The ΔLAI in June continues to have a positive lagging effect on the GPP. For DBF, although the impact of ΔLAI on the GPP in April and May was always positive, this effect weakened in June. For DNF, the increase in ΔLAI in April promoted GPP in the months before October. ΔLAI and GPP in May and June showed similar positive correlations.

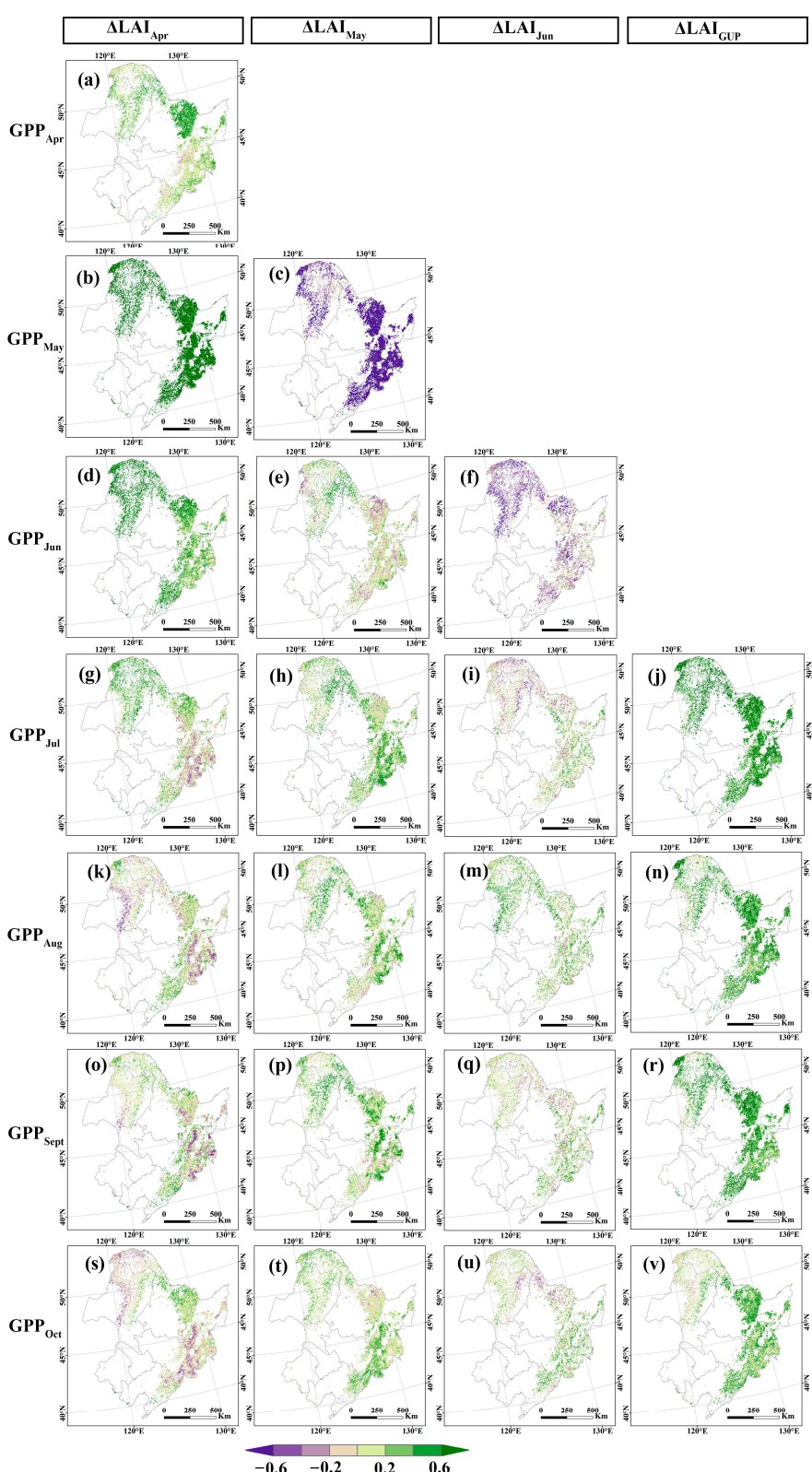

**Figure 8.** Spatial distribution of Pearson correlation coefficients between the entire leaf green-up period (GUP) and monthly ΔLAI and GPP from April to October in forests in northeast China for the period 2001–2021. (**j,n,r,v**) Entire GUP (ΔLAI is defined as the difference between the annual maximum LAI and LAI one month before the beginning of the growing season), (**a,b,d,g,k,o,s**) April, (**c,e,h,l,p,t**) May and (**f,i,m,q,u**) June (monthly ΔLAI is defined as the difference between the LAI of the current month and the LAI of the previous month) and GPP from April to October, respectively.

## 4. Discussion

### 4.1. Spatiotemporal Distribution of Leaf Carbon Distribution

This study provided the first large-scale explanation of forest carbon allocation to leaves and accumulation. The study found that ΔLAI of almost all forest types reached the highest in April and decreased in May compared to that in April (Figure 2), probably because the critical period for leaf development in most forestlands is April, and almost all the carbon income during this period is used for leaf growth. By May, when the leaves are fully developed, the ΔLAI decreases [26]. However, in some areas of the MF, the value was low in April, and it reached its maximum value in May. As the study area belongs to a nature reserve with almost no impact from human activities, there may be two reasons for this occurrence in the MF. First, the unique and complex community attributes of the MF may cause the late sprouting of leaves [27]. Second, this part of the forest experienced a drought in April. It is common for forests to delay the development of leaves and reduce transpiration water loss caused by increased leaf area during a drought [28,29]; however, the growth rhythm of the forests was irreversible. Therefore, after stress is relieved in May, forests first use as much carbon income as possible for leaf growth [30]. As an example, in 2005, northern China experienced a severe spring drought, and hence, the ΔLAI in April was significantly lower than the multi-year average, while the ΔLAI in May was higher than the multi-year average. The optimal allocation theory states that, after environmental stress, a tree adjusts the proportion of carbon allocation in different organs [31] and allocates more carbon to the growth of stilts to promote the long-term survival of the plant [32]. Therefore, the delayed leaf growth strategy will have an irreversible impact on the GPP throughout the year [33].

Multiyear trends of carbon allocation to leaves can characterize the health of ecosystems [34]. If the carbon absorbed by photosynthesis is allocated to other long-term tissues after the forests complete canopy growth each year, the ΔLAI in April will increase each year and decrease in May each year (Figure 4b,c). This shows that, in areas where this trend exists, the main stage of leaf growth of vegetation from 2001 to 2020 was increasingly concentrated in April. This phenomenon ensures a longer time for trees to use photosynthesis for the growth and development of organs, such as xylem and roots, and thus fixes more carbon into non-photosynthetic organs. Therefore, this phenomenon enhances the capacity of the forest ecosystem to sink carbon, gradually improves its structure and function, and establishes a solid ecological protection barrier.

### 4.2. Driving Factors of Leaf Carbon Distribution

This study used RF and SEM methods to explore the complex driving factors of carbon allocation to leaves from different perspectives. Significant differences in the main drivers of carbon allocation to leaves in different forests were observed [35]. TEM had a stronger positive effect on the increase in ΔLAI in the early stages (April) of DBF and MF. No obvious difference in the degree of influence of meteorological factors on DNF was observed (Figures 6 and 7). In addition to the special growth properties of DNF, their spatial distribution can also cause this phenomenon. The DNF is located above 50° N and leaf development does not begin in April, whereas the areas of DBF and MF are generally located further south. Although leaves begin to grow in April, leaf growth at this stage is still mainly limited by TEM [36]. Carbon allocation to leaves in DBF and MF was mainly driven by TEM until May, but the degree of influence was significantly lower than that in April. However, for DNF, which begins leaf growth only in May, PRE was the dominant factor. This phenomenon can be attributed to the properties of different forest types [37]. DBF and MF have a larger special leaf area and poor cold tolerance and the leaf development in them is dominated by TEM, whereas the leaves of DNF are resistant to cold. Therefore, leaf development is determined by PRE in DNF [38]. The development of leaves in DNF will be completed by June, and the effects of all meteorological factors will be minimized. Although DBF and MF exhibit the same decreasing pattern, their importance

indexes are higher than that of DNF because of their higher sensitivity to meteorological factors than DNF [39].

In addition, SOS has high path coefficients for ΔLAI in different time periods. Spring phenology has the most significant impact on carbon allocation by changing the biophysical feedback and other seasonal biological processes [40]. Among all influencing factors, SOS had the greatest impact on carbon allocation to leaves at all stages and for all tree species as it is a natural phenomenon manifested by trees under the influence of multiple and complex environmental conditions. The influence of SOS on leaf carbon distribution is more comprehensive and representative than those of other factors [41]. We found that SOS has a negative path coefficient for ΔLAI in April (Figure 7), which indicates that an earlier SOS will promote the growth of ΔLAI in April because an earlier spring phenology enhances the growth of spring vegetation [42], and results in a stronger positive impact on subsequent forests growth, thereby promoting leaf growth in April. However, there are positive path coefficients for the entire GUP and leaf growth in May and June, indicating that early SOS will be detrimental to GUP and ΔLAI growth in May and June (Figure 7), which may be due to the earlier spring phenology. A longer growing season enhances spring forest growth, and therefore, an increase in evaporation and reduction in soil moisture will lead to water stress in the later stages of growth [43,44], During this period, forest growth is the most sensitive to water availability; ultimately, biophysical processes caused by the early spring phenology will negatively impact the water supply in summer.

### 4.3. Effects of Carbon Allocation to Leaves on Gross Primary Productivity

There is a positive correlation between ΔLAI and GPP in almost all forest types from April to June (Figure 8), indicating that the development of leaves can directly affect forest productivity. In addition, because of the implementation of various ecological restoration projects from 2001 to 2021, the development status of forest canopy leaves and their photosynthetic carbon sequestration capacity have displayed an upward trend through the years [45], resulting in a positive correlation between ΔLAI and GPP. There are two main reasons for this situation. The first is the driving effect of photosynthesis. Leaves are the main site for plant photosynthesis, which is the process of energy conversion and carbon fixation in plants. Therefore, a larger leaf area provides more sites for photosynthesis and more energy and carbon sources for plants, thus promoting the total primary productivity of the plants [46]. The second reason is the absorption of nutrients and water. Better leaf development increases the root surface area available for the absorption of water and nutrients. Therefore, it enables trees to acquire water and nutrients more efficiently to support an increase in their total primary productivity [47].

The leaf development in a given month has a time requirement for the promotion of GPP. For example, ΔLAI in April does not directly increase GPP in April. It has the strongest positive correlation with GPP in May and is also positively correlated with GPP in other months, but the correlation gradually weakens. Similarly, the increase in ΔLAI in May and June is more conducive to an increase in GPP in the later part of the growing season. The relationship between ΔLAI and GPP is the result of the combined effects of internal physiological regulation of trees and external environmental influences. First, April is still within the transition season, when the weather is cooler, there is less sunlight, and the humidity is high. Therefore, although leaf area increases, there may not be a direct positive correlation with GPP in April [48]. Second, May is usually the month when spring transitions to early summer, which provides sufficient light energy and suitable temperatures, allowing plants to utilize leaf area for photosynthesis more effectively [49]. Finally, as the growing season progresses, during May and June, trees may have fully entered the growing season stage with greater growth and photosynthetic capabilities. At this time, plants can better utilize an increase in leaf area to enhance photosynthesis and carbon fixation capacity, so the increase in ΔLAI in May and June is more conducive to an increase in GPP in the later part of the growing season [50].

However, the impact of ΔLAI on the GPP is not always positive, and the opposite phenomenon occurs for individual forest types at specific growth stages. Especially in MF, the increase in ΔLAI in April has a positive effect on the GPP from April to June but becomes an inhibitory effect on the GPP in July and subsequent months. This phenomenon is caused by the overdevelopment of the canopy in the early part of the growing season, thereby promoting the initial GPP; however, too many leaves lead to an increase in water transpiration [51]. If the water supply in the environment is limited, excessive leaves may intensify the plant's water transpiration, causing it to face the risk of dehydration and water stress, which are not conducive to a later increase in GPP [52]. In addition, the increase in the number of leaves leads to the requirement of a high quantity of soil moisture and nutrients [53], which is not conducive to carbon and water synthesis in the later stage of photosynthesis [54].

*4.4. Limitations and Prospects*

This study explains the drivers of carbon allocation to leaves and their impact on the GPP in the boreal forests of China. However, the method used for the analysis of temporal changes using ΔLAI as a proxy for leaf carbon has uncertainties. The cessation of the increase in ΔLAI does not imply that the carbon allocation to the leaves will not increase. The carbon consumed by leaf respiration is also a part of the carbon allocation to leaves. Therefore, future studies should combine experimental data from field operations with long-term observational data on plant carbon allocation. Furthermore, our analysis was performed at a spatial resolution of $0.05° \times 0.05°$, which is coarse and may lead to considerable uncertainty [55]. For example, for mixed pixels in the forest-grass boundary zone, changes in LAI may also be caused by changes in forest type rather than changes in climate or phenology. Therefore, additional influencing factors must be considered to explain the true driving mechanisms of forest carbon allocation to leaves. Although remote sensing-based methods provide new possibilities for studying leaf carbon allocation, the lack of long-term and large-scale field measurement datasets of leaf biomass attributable to the large scope of the study area will continue to prevent us from exploring long-term changes in leaf carbon allocation for many years. Thus, the progress of future research on ecosystem leaf C allocation may still be limited by the research basis of plant trait observation networks.

**5. Conclusions**

This paper studies the spatiotemporal distribution and main driving mechanisms of carbon allocation to leaves (ΔLAI) in forests in northeast China at five different time scales (GUP, April, May, June, and July), and elucidates the differences in the impact of ΔLAI on the GPP. The following conclusions were drawn:

(1)  Owing to the differences in physiological attributes, in the GUP, the ΔLAI values of DBF and MF are much higher than that of DNF, and all three show an insignificant increasing trend each year. The highest ΔLAI in DBF occurred in April and in DNF and MF it occurred in May. The ΔLAI of DBF and MF showed a significant year-by-year increasing trend in April, and DNF showed a significant increasing trend in most areas in May;

(2)  The main factors driving ΔLAI in GUP are TEM and SOS. The main driving factors in April and May were SR and SOS. The driving mechanism in June was the most complex, and the difference between different forestlands was the highest. Except for PRE in DBF and MF, all other factors had larger path coefficients. The coefficients of SR and SOS were the highest for DNF;

(3)  ΔLAI in GUP has a significant impact on the GPP. In the MF, the higher ΔLAI in May was most conducive to an increase in GPP. In DBF and DNF, the ΔLAI in April and May both promote the increase of GPP.

This study examined the changes in forest carbon allocation to leaves and its driving factors on a large scale, providing an important basis for understanding the combined

influence of plant traits and the environment in restricting carbon allocation strategies. It is evident that the development of forest leaves is based on environmental factors and attributes of forestlands. During the critical period, that is April, carbon allocation to leaves effectively improves the carbon sequestration capacity of forestland. This information is of great value for the development and validation of terrestrial ecosystem models.

**Author Contributions:** Conceptualization, methodology, and writing—original draft, Z.L.; methodology, supervision, and inspection Q.L.; software Z.L. and Q.L.; validation, Y.B.; review and editing, B.S.; investigation and validation, Z.B.; Formal analysis, X.L. All authors have read and agreed to the published version of the manuscript.

**Funding:** This research was supported by the Inner Mongolia Autonomous Region Natural Science Foundation (2022MS04006 and 2022LHQN04002) and the Introduction of High-Level Talents Scientific Research Start-up Fund Project (2022JBYJ030).

**Data Availability Statement:** Dataset available on request from the authors.

**Acknowledgments:** The authors thank the editors and the anonymous reviewers for their crucial comments, which improved the quality of this paper.

**Conflicts of Interest:** The authors declare no conflict of interest.

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
