# Peer review of "Carbon Allocation to Leaves and Its Controlling Factors and Impacts on Gross Primary Productivity in Forest Ecosystems of Northeast China"

_forests, doi:10.3390/f15010129_

Round 1
Reviewer 1 Report
Comments and Suggestions for Authors
Overall the introduction is satisfactory and justifies sufficient background study as per the title. However, in the second last paragraph line 70-77, “Some studies have found that trees adapt to environmental stress through dynamic 70 physiological adjustment mechanisms, thereby reducing negative impacts and maximiz- 71 ing carbon sequestration [8]. Gross primary productivity (GPP) is a vital indicator of eco- 72 system productivity. However, the extent of the effects of changes in carbon allocation to 73 leaves remain unknown. Current research on the factors affecting GPP mainly considers 74 the role of climatic factors, whereas the indirect impact of forest adaptation strategies (car- 75 bon allocation) on the ecosystem has been insufficiently studied, although this adjustment 76 strategy within trees may be greater than the direct impact of climate [14].” the authors should need modify this paragraph about the Gross primary productivity (GPP).
Clarify the aims of the study at end of the introduction.
Figure 1a is blurred would be better to provide good quality.
Figure 2: The x- and y-axis labels are not clear please provide clear images.
Figure 5 same as above.
Figure 7: Not visible. Please provide a good quality.
Overall, The figure's quality is not satisfactory.
Line 351-352: “The ΔLAI in April has the strongest positive correlation with GPP in May, and the lag time is one month”. In order to perform Pearson correlation, so it must have r-value, r2, p, t and df? Right?
Comments on the Quality of English LanguageMinor editing of English language required
Reviewer 2 Report
Comments and Suggestions for Authors
The manuscript entitled “Carbon allocation to leaves and its controlling factors, and impacts on gross primary productivity in forest ecosystems of Northeast China" is scientifically correct. The topic is important for forests ecosystems. The information generated form the research can be defined as valuable for the development and validation of terrestrial ecosystem models.
The structure of the manuscript is well organized. The authors have processed a big amount of data and presented it in an extremely good manner.
I have some minor comments and suggestions with view to improve much more the quality of the manuscript.
- Maybe it is good to remove the comma from the title of the manuscript. If the comma removal will change authors’idea then please consider another way to express what you mean in the title.
- In section 4.4 Shortcomings and prospects the authors mentioned “uncertainties”. Please consider possibility to develop further the issue with uncertainties and go further and with more details.
- Please consider deeply and more clearly the connection between ΔLAI and GPP and also additional information/analysis/conclusions on the impacts on GPP in forest ecosystems will be valuable. (maybe in section 4.3 Effects of carbon allocation to leaves on gross primary productivity , 5. Conclusions or other appropriate place according to the authors).
Good luck,
Reviewer#
